# Screening for Mild Cognitive Impairment in the Preoperative Setting: A Narrative Review

**DOI:** 10.3390/healthcare10061112

**Published:** 2022-06-15

**Authors:** Mariska te Pas, Marcel Olde Rikkert, Arthur Bouwman, Roy Kessels, Marc Buise

**Affiliations:** 1Department of Anesthesiology, Catharina Hospital, 5623 EJ Eindhoven, The Netherlands; arthur.bouwman@catharinaziekenhuis.nl (A.B.); marc.buise@catharinaziekenhuis.nl (M.B.); 2Radboud University Medical Center, Department of Geriatric Medicine, 6500 GL Nijmegen, The Netherlands; marcel.olderikkert@radboudumc.nl; 3Department of Electrical Engineering, Eindhoven University of Technology, 5612 AZ Eindhoven, The Netherlands; 4Donders Institute for Brain, Cognition and Behaviour, Radboud University, 6525 XZ Nijmegen, The Netherlands; roy.kessels@donders.ru.nl; 5Department of Medical Psychology, Radboud University Medical Center, 6525 GA Nijmegen, The Netherlands; 6Vincent van Gogh Institute for Psychiatry, 5803 AC Venray, The Netherlands

**Keywords:** mild cognitive impairment, preoperative, cognitive screening, sensitivity and specificity, postoperative cognitive dysfunction (POCD), postoperative delirium (POD)

## Abstract

Cognitive impairment predisposes patients to the development of delirium and postoperative cognitive dysfunction. In particular, in older patients, the adverse sequelae of cognitive decline in the perioperative period may contribute to adverse outcomes after surgical procedures. Subtle signs of cognitive impairment are often not previously diagnosed. Therefore, the aim of this review is to describe the available cognitive screeners suitable for preoperative screening and their psychometric properties for identifying mild cognitive impairment, as preoperative workup may improve perioperative care for patients at risk for postoperative cognitive dysfunction. Electronic systematic and snowball searches of PubMed, PsycInfo, ClinicalKey, and ScienceDirect were conducted for the period 2015–2020. Major inclusion criteria for articles included those that discussed a screener that included the cognitive domain ‘memory’, that had a duration time of less than 15 min, and that reported sensitivity and specificity to detect mild cognitive impairment. Studies about informant-based screeners were excluded. We provided an overview of the characteristics of the cognitive screener, such as interrater and test-retest reliability correlations, sensitivity and specificity for mild cognitive impairment and cognitive impairment, and duration of the screener and cutoff points. Of the 4775 identified titles, 3222 were excluded from further analysis because they were published prior to 2015. One thousand four hundred and forty-eight titles did not fulfill the inclusion criteria. All abstracts of 52 studies on 45 screeners were examined of which 10 met the inclusion criteria. For these 10 screeners, a further snowball search was performed to obtain related studies, resulting in 20 articles. Screeners included in this review were the Mini-Cog, MoCA, O3DY, AD8, SAGE, SLUMS, TICS(-M), QMCI, MMSE2, and Mini-ACE. The sensitivity and specificity range to detect MCI in an older population is the highest for the MoCA, with a sensitivity range of 81–93% and a specificity range of 74–89%. The MoCA, with the highest combination of sensitivity and specificity, is a feasible and valid routine screening of pre-surgical cognitive function. This warrants further implementation and validation studies in surgical pathways with a large proportion of older patients.

## 1. Background

In the elderly population, postoperative delirium and cognitive dysfunction are both associated with poor postoperative outcomes, such as decreased quality of life, higher health care costs, and increased mortality [1]. According to the World Health Organization, the proportion of the world’s population over 60 years will nearly double from 12% to 22% between 2015 and 2050 [2]. Parallel to this, the number of elective surgeries in older patients is rising [3]. Therefore, older patients are a growing group with varied vulnerability who increasingly undergo surgical procedures.

Delirium is an acute decline in cognitive functioning that is a common, serious, and potentially fatal complication affecting up to 50% of hospitalized seniors and is costing over $164 billion (2011) per year in the United States and over $182 billion (2011) per year in 18 European countries [4]. Delirium is defined as an acutely and fluctuating state of reduced attention or even reduced consciousness and cognitive decline that is triggered by physical stressors and/or medications. In the general hospital setting, delirium remains a common, serious, and under-recognized problem affecting mostly older patients [5,6,7].

In 1887, George Savage [8] described how surgery and anesthesia may possibly contribute to the development of “mental insanity” [9]. The term Postoperative Cognitive Dysfunction (POCD) was first described by Bedford in 1955 in an article called ‘Adverse cerebral effects of anesthesia on old people’ [10]. Currently, POCD is defined as an impairment of cognitive function arising after a surgical procedure. POCD refers to disorders affecting orientation, attention, perception, and judgment that develop after surgery [11]. The prospective longitudinal study of Monk and colleagues showed that at hospital discharge, POCD was present in 36.6% young, 30.4% middle-aged, and 41.4% older adults [12].

It is important to identify patients at risk, as signs of early cognitive impairment can be subtle and often have not been previously diagnosed yet will raise the risk of developing delirium after surgery [13]. This is even recommended by the fifth international perioperative neurotoxicity working group, with over 30 experts included who developed recommendations specific to postoperative brain health in individuals >65 years of age [14]. Early stages of cognitive decline are described by a variety of terms, such as amnestic mild cognitive impairment, mild neurocognitive impairment, dementia prodrome, incipient dementia, or isolated memory impairment. In the Diagnostic and Statistical Manual of Mental Disorders–Fifth Edition (DSM–V, the construct is classified as mild neurocognitive disorder [15,16]. In this review, we limit it formally to mild cognitive impairment (MCI) described along a continuum between normal aging and dementia [17] with criteria defined by Petersen [18], as patients with MCI have a higher likelihood of developing perioperative delirium and post-surgery further cognitive decline than patients without cognitive impairment [19].

Identifying MCI at the preoperative outpatient clinic is essential for risk stratification of POCD and postoperative delirium [20,21]. The longer a delirium remains untreated, the more cognitive decline may be expected [22]. It is therefore important to identify patients at risk to start early etiology-based and symptom-based treatment [7]. Although there is no established prophylaxis or treatment of POCD itself, it is important that the population at risk can be clearly identified to offer better perioperative care. In 2015, Saleh et al. found that cognitive training reduces the decline of early postoperative cognitive function in elderly patients [23]. For methods assessing cognitive function to be feasible in the preoperative workup, these tools should best be administered in or by the patient (i.e., not based on informant information), easy to administer, short, and attractive [24], yet valid and reliable from a psychometric perspective [25].

At present, the most frequently used and investigated cognitive assessment tool is the Mini-Mental State Examination (MMSE) [26]. Although the MMSE is widely used, it has been criticized with respect to its psychometric properties [27] and has limited sensitivity for detecting mild cognitive impairment (MCI) [28]. In addition, the administration time of the MMSE may sometimes exceed 15 min [29], which makes it not optimal for use as a quick screening tool in the preoperative setting [29]. Furthermore, the efficacy of routine use of cognitive screening tools in perioperative patients has not been fully established [30,31]. Therefore, currently, preoperative cognitive screening to evaluate baseline cognitive functioning is not routinely performed in the preoperative assessment.

The aim of this paper is to provide an overview of screeners currently available which are efficient (≤15 min) and suitable in the preoperative setting and to summarize the existing literature on the diagnostic value of these screeners for recognizing mild cognitive impairment (MCI) in older patients. The diagnosis of MCI was preferably made according to Petersen’s criteria [18].

## 2. Methods

Electronic searches of PubMed, PsycInfo, ClinicalKey, and ScienceDirect were conducted to identify relevant literature about appropriate screeners published in the period 2015–2020. This period was selected to limit the first part of the search to recent validity research on these screeners. Only manuscripts published in English were included in the review. The following search terms were used: Mental status and dementia tests (MeSH), cognitive screening, cognitive screeners, screener, mass screening (MeSH), screening tool, assessment, test, cognitive impairment, mild cognitive impairment, MCI, cognitive dysfunction (MeSH), cognitively impaired, aged (MeSH), elderly, geriatrics, older, elderly, elderly people, old people, and senior. The flowchart of the search is shown in Figure 1.

We defined inclusion criteria to select journal articles and reviews which discuss cognitive screeners with appropriate properties to detect MCI at the preoperative outpatient clinic. From the journal articles and reviews, we extracted the articles including screeners that (1) cover at least the cognitive domain ‘memory’, because amnestic MCI is the subtype that most often develops Alzheimer’s dementia, which has the highest prevalence of the dementia types [32,33]. Memory deficits are both the most common complaint in MCI and the cardinal feature of Alzheimer’s dementia. Furthermore, complaints about poor memory are the most frequent reason for referral to a memory clinic and provide a good starting point for the consultation. The diagnosis of MCI is established by evidence of memory impairment [34]. Besides that the screener should cover the cognitive domain memory, the screener should (2) have a mean administration time of 15 min or less, which is an arbitrary limit, but one that has been used in previous literature [35]. While the time saving of about 1–2 min may not seem remarkable for an individual end-user, it is impactful in a health service with a high patient volume and limited manpower [36]. Next, the screener (3) had to be validated against a comprehensive neuropsychological assessment as the gold standard, and (4) have documented the sensitivity and specificity for Mild Cognitive Impairment (ideally in the elderly surgical population). The internet was searched to find a mean test administration time if this had not been described in the journal article or review.

Studies found by the electronic search were excluded (1) if there was inadequate reporting (e.g., if studies fail to report the screener’s sensitivity or specificity of the screener for mild cognitive impairment), (2) if the screener did not assess at least the cognitive domain memory, (3) if the screener was only studied for cognition assessment in a specific nonsurgical disease group (e.g., depressed patients, brain tumors, etc.) because this is not representative for our target group, (4) if only participants with diagnosed dementia were included, as this is not representative for our target group, (5) if the screener was informant-based, as the suitability of informant questionnaires for the assessment of MCI is uncertain [37], (6) if any test performance bias occurred (incorporation bias, verification bias, diagnostic review bias, or test review bias), and (7) if the paper was published before 2015. If the first reviewer was in doubt about inclusion, a second reviewer was consulted, and together they discussed the pros and cons until reaching a consensus.

After the systematic search and identification of 10 suitable screeners, we performed a snowball search to obtain further literature about these 10 screeners. For this snowball search, we searched the aforementioned databases for every screener separately. In addition, we aimed to trace the original article about the development and validation of the screener. From there on, we searched the reference list for relevant articles providing more information about the diagnostic properties of the screener. The aforementioned inclusion and exclusion criteria were applicable to this snowball search. For the snowball search, there was no predefined period chosen, as we wanted to include original articles. In the end, although we started with the identification of papers published between 2015 and 2020, articles published before 2015 could also be included in this review. Unless described otherwise, MCI was diagnosed according to Petersen’s criteria [18].

For data extraction, we used the STARDdem criteria as a guideline to determine the most important criteria for the effective evaluation of studies for diagnostic tests in dementia and cognitive impairment [38]. The criteria used in this review are the identification of articles describing sensitivity and specificity, participant sampling, index test and reference test description, the definition of cutoff scores, presence of bias, and reliability. These criteria are summarized in the tables in the results section.

## 3. Results

The systematic search executed as described above resulted in 4775 titles. A total of 3222 titles were excluded from the analysis because the papers were published before 2015. A total of 1448 studies were excluded as they did not focus especially on older people or focused on older people who were already diagnosed with Alzheimer’s dementia. In total, fifty-three titles did not fulfill the inclusion criteria, as most screening tools did not measure general cognitive functioning or they focused on cognitive functioning in a specified disease, such as depression or multiple sclerosis. All abstracts of the remaining 52 studies were examined and resulted in 46 different screeners. Of these 46 screeners, 10 met the inclusion criteria. For these nine screeners, a further snowball literature search was executed to obtain more evidence of their diagnostic properties. Figure 1 shows the flowchart of the search process. Reasons for excluding screeners are displayed in Table 1. Table 2 shows a summary of the various cognitive screeners according to the STARDdem criteria [38]. The test-retest reliability and the interrater reliability correlations are displayed in Table 3. Not all forms of reliability were traceable in the literature for all screeners. We will subsequently describe the most relevant findings for the included screeners. Screeners included in this review are the Mini-Cog, MoCA, O3DY, AD8, SAGE, SLUMS, TICS(-M), QMCI, MMSE2, and Mini-ACE.

## 4. Mini-Cog

The Mini-Cog was developed in approximately 2000 to identify individuals with clinically important cognitive impairment in nonspecialist settings [69]. The Mini-Cog consists of a three-word recall test and a clock drawing test, and it takes 3 min to complete. The scoring system is as follows: 1, 2, or 3 points for a 3-item recall score, 2 points for drawing a normal clock, and 0 points for drawing an abnormal clock. The advice is that a total score of 3 or more points of the maximum of 5 points indicates a lower likelihood of dementia but does not rule out some degree of cognitive impairment [70].

The developmental study of the Mini-Cog and a subsequent study by Borson and colleagues excluded patients diagnosed with MCI and focused mainly on the sensitivity and specificity of the Mini-Cog for dementia [69,71].

A study by Carnero-Pardo and colleagues [39] pooled the data from two studies and defined cognitive impairment as mild cognitive impairment + dementia. The cognitive diagnosis was performed by two senior neurologists with expertise in cognitive and behavioral neurology after a medical visit, a neurological exam, a mental status exam, and a formal neuropsychological evaluation, which included tests of memory, attention/executive functions, language, and visual-spatial abilities. This study reported a sensitivity of 60% and a specificity of 90% for the Mini-Cog to detect overall cognitive impairment, i.e., the sum of MCI and Alzheimer’s Disease. The optimal cutoff score was 2 or below.

Another study by Li and colleagues [40] reported that the Mini-Cog had a sensitivity of 85.71% and a specificity of 79.41% for detecting mild cognitive impairment. The patients were divided into an MCI group (*n* = 119) and a non-MCI (*n* = 110) group based on the final evaluation by a neurologist.

## 5. Montreal Cognitive Assessment (MoCA)

The Montreal Cognitive Assessment (MoCA) is a brief, 30-question test that helps healthcare professionals detect subtle signs of cognitive impairment, allowing for early diagnosis and patient care [72]. It takes 10 min to complete. The MoCA was first published by Nasreddine and colleagues, followed by a validation study that reported a sensitivity of 90% and a specificity of 87% for detecting MCI at a cutoff point of 25 or below [41]. Patients were diagnosed in a memory clinic supported by a neuropsychological evaluation.

Freitas and colleagues reported a sensitivity of 81% and a specificity of 77% for detecting MCI at an optimal cutoff point of 21 or below. The diagnosis was established by a multidisciplinary team based on consensus considering the results of the comprehensive assessment [42]. The reliability of the Polish version of the MoCA 7.2 was analyzed by Sokolowska and colleagues, who demonstrated that the MoCA detects MCI with a sensitivity of 89.5% and a specificity of 74.1 at a cutoff point of 24 or below. In this study, the recommended cutoff point for the MoCA resulted in a sensitivity of 43.3% [43]. Identification of MCI was based on the DSM-5 criteria [73]. A study by Fujiwara and colleagues on the Japanese MoCA reported a sensitivity of 93% and a specificity of 89% for detecting MCI at a cutoff point of 25 or below. All patients with MCI were seen at the Memory Clinic of Tokyo Metropolitan Geriatric Hospital in Tokyo [44].

## 6. Ottawa 3 Day Year (O3DY)

The Ottawa 3 Day-Year is a four-item screening instrument. Patients are asked to state the day, date, and year and to spell WORLD backward as DLROW. Molnar and colleagues formed this new test based on 12 neuropsychological tests, background information, and established normative information. The main goal in creating this screening tool was simplicity. For that reason, it does not include questions that required the use of paper and pen, cue cards, or props, with no question requiring more than 30 s to answer. In the validation study, a sensitivity of 80% and a specificity of 61% were reported with a cutoff score of 2 or more. All subjects, participants of the Canadian Study of Health and Aging [74], and controls received a full clinical evaluation that included extensive medical, neurological, and neuropsychological examinations to form two groups, namely, cognitively impaired and cognitively normal subjects [45].

No other evidence is available on the validity of the O3DY for the detection of mild cognitive impairment by using a comprehensive neuropsychological assessment.

## 7. The 8-Item Interview to Ascertain Dementia (AD8)

The Eight-item Interview to Differentiate Aging and Dementia questionnaire (AD8) consists of eight yes/no questions that are administered by the participant without the assistance of medical or research staff. It typically takes 2–3 min to complete. The AD8 targets the participant’s perception of his or her memory, problem-solving abilities, orientation, and the impact of cognitive function on daily activities over the past several years. It is a screening test for very mild cognitive impairment. The eight items are added up, resulting in a score ranging from 0 to 8. The suggested cutoff score for impaired cognition is 2 or higher [75].

Galvin and colleagues investigated the sensitivity and specificity of informant-based AD8 to detect very mild dementia described as a clinical dementia rating (CDR) of 0.5 [34]. The most desirable combination of sensitivity (74%) and specificity (86%) was achieved with a cutoff score of 2 to predict a CDR ≥ 0.5 [46].

Although, as mentioned before, the AD8 was originally developed and validated as an informant-based measure and should therefore be excluded, recent studies validated the AD8 as a questionnaire for people with potentially impaired cognitive functions [76].

Another study by Galvin and colleagues reported that the participant’s self-rating had the best combination of sensitivity (80%) and specificity (59%) using a cutoff score of 1 for a CDR ≥ 0.5. An independent psychometric battery was administered to all participants, but the clinician was blinded to the results [47].

## 8. Self-Administered Gerocognitive Examination (SAGE)

The SAGE is a 12-item, self-administered examination for detecting MCI and early dementia in elderly patients. The SAGE was developed based on cognitive testing and research. It requires no equipment or personnel to administer and tests several cognitive domains: orientation to date, language, memory, executive function, visuospatial abilities, and calculation. The mean administration time is 15 min [48].

Scharre and colleagues noted a sensitivity of 79% and a specificity of 95% in detecting overall cognitive impairment, which means mild cognitive impairment + dementia, at a cutoff score of 16 or below. ROC analysis of SAGE in cognitively unimpaired individuals versus MCI patients showed a sensitivity of 62% and a specificity of 95% [18]. The clinical evaluation included a detailed medical history, physical and neurologic examination, and a neuropsychologic battery [48].

Another study by Scharre and colleagues investigated the characteristics and utility of SAGE as an online cognitive screening test, classifying 66 subjects as either dementia, MCI, or normal based on standard clinical criteria and on their neuropsychological test scores. A score of SAGE at or below 15 provided 71% sensitivity in detecting cognitive impairment and a specificity of 90%. A score of SAGE at or below 16 provided 69% sensitivity in detecting cognitive impairment and a specificity of 86% [49].

## 9. Saint Louis University Mental Status Examination (SLUMS)

The SLUMS was developed to address limitations of the MMSE, especially with regard to more educated patients, and as a screening tool for mild neurocognitive disorder. The SLUMS consists of 11 items and measures aspects of cognition that include orientation, short-term memory, calculations, the naming of animals, the clock drawing test, and recognition of geometric figures. Scores range from 0 to 30, with a score between 21 and 26 indicating mild cognitive impairment. Severe cognitive impairment was defined as a score of ≤20 points. It takes approximately 7 min to administer SLUMS [15].

A study by Tariq and colleagues used the DSM-IV [77] criteria to establish the diagnosis of mild neurocognitive disorder or dementia. Participants were evaluated during a routine clinical visit, and their histories were obtained from corroborating sources. A complete physical and mental status examination was performed. The optimal cutoff scores for SLUMS for mild cognitive impairment in patients with less than high school education and in patients with high school or higher education were 23.5 and 25.5, respectively. At a cutoff of 23.5, a sensitivity of 92% and a specificity of 81% were reported. At a cutoff of 25.5, a sensitivity of 95% and a specificity of 76% were reported [15].

Another cutoff score was described in the study of Shwartz and colleagues. The optimal cutoff score for differentiating patients with MCI from patients with no diagnosis was slightly lower than the cutoff score reported by Tariq and colleagues. Specifically, a score of 25 or less was suggestive of a diagnosis of MCI with a sensitivity of 81% and a specificity of 68%. A comprehensive neuropsychological battery was administered to support the MCI diagnosis [50].

## 10. Telephone Interview Cognitive Status (Modified) (TICS(-M))

The TICS was originally developed as an 11-item screener with a maximum total score of 41 [78]. The TICS has some similarities to the Mini-Mental Status Examination in that it includes questions regarding orientation, repetition, and naming. However, to increase the probability of identifying cognitive impairment, the TICS was subsequently modified (TICS-M) to include a more comprehensive memory assessment, including both immediate and delayed recall of a 10-item list of nonsemantically related words. The TICS-M consists of 13 items with a maximum total score of 50 [51].

The study of Cook and colleagues sought to determine the sensitivity and specificity of detecting amnestic MCI with the TICS-M within a non-clinical community-based volunteer sample. Participants individually underwent a 2.5-h multidomain neuropsychological battery. Together with a CDR score, the level of cognitive functioning was determined. A CDR of 0.5 was indicative of mild cognitive impairment. At a cutoff score of 34, a sensitivity of 82.4% and a specificity of 87.0% were described [51].

Knopman and colleagues merged patients with MCI with people with dementia. A CDR of 1 or higher was indicative of dementia. Independent of the administration of the TICS-M by telephone, all participants underwent a full in-person assessment that included a neuropsychological test battery, a physician examination, and an interview with someone close to the participant. ROC analyses identified a score of 31 or lower as the optimal cutoff score to separate subjects with MCI from normal subjects. For detecting MCI at this cutoff score, a sensitivity of 71.4% and a specificity of 78.3% were reported. For detecting cognitive impairment (MCI + dementia) at the same cutoff score of 31, the sensitivity was 83.3%, and the specificity was 78.3% [52].

## 11. Quick-MCI (QMCI)

The Quick Mild Cognitive Impairment Screen (QMCI) was developed by O’Caoimh and colleagues [79] with the purpose of creating both a quick and a more accurate screening test for detecting MCI and dementia. The QMCI has a maximum total score of 100 and comprises six subtests: orientation (10 points), registration (5 points), clock drawing (15 points), delayed recall (20 points), verbal fluency (20 points), and logical memory (30 points). The QMCI can be completed in 3 to 5 min.

O’Caoimh and colleagues reported a sensitivity of 82% and a specificity of 70% for detecting MCI and a sensitivity of 91% and a specificity of 80% for detecting cognitive impairment in subjects attending four memory clinics across Ontario, Canada, referred for the investigation of cognitive loss between 2004 and 2010. A diagnosis of dementia was based on NINCDS [80] and DSM-IV [77] criteria. A diagnosis of MCI was made by a consultant geriatrician if patients had recent, subjective but corroborated memory loss without obvious loss of social or occupational function [53].

Bunt and colleagues adapted the QMCI for use in Dutch-language countries and studied the validity of the Dutch version of the QMCI (QMCI-D). The MCI and dementia groups underwent the same comprehensive review at baseline, including neuropsychological assessment and magnetic resonance imaging or computerized tomogram. The QMCI-D had a sensitivity of 70% and a specificity of 94% in differentiating MCI subjects from controls at a cutoff score of 51.5. For differentiating cognitive impairment from controls, a sensitivity of 82% and a specificity of 90% were reported [54].

In a review by Glynn and colleagues, the QMCI was compared to other short cognitive screening tests. In total, seven articles were included comparing the QMCI with other cognitive screens in terms of accuracy, sensitivity, and specificity in detecting MCI and dementia. All articles used groups of individuals diagnosed with normal cognition, MCI, or dementia, with screening tests administered by raters who were blinded to the diagnosis. Pooled data from these seven articles showed a sensitivity of 82% and a specificity of 82% for differentiating mild cognitive impairment from normal cognition. When differentiating cognitive impairment from normal cognition, a sensitivity of 95% and a specificity of 83% were reported [55].

## 12. Mini-Mental State Examination, 2nd Edition (MMSE-2)

The Standard Version of the MMSE-2 (MMSE-2:SV) is one of the three revised versions of the MMSE. It retains the structure and scoring of the original 30-point MMSE, but some items were replaced, and several tasks were modified to minimize the difficulty and facilitate its translation into foreign languages. In addition to the SV, there is also a Brief Version (BV) and an Expanded Version (EV). Because of the administration time of 20 min for EVs, the expanded version was excluded from this review. The SV and the BV have administration times of 10–15 and 5 min, respectively [81]. The MMSE-2:BV is composed of four subtests in the following order: registration, orientation to time, orientation to place, and recall. The MMSE-2: SV is composed of seven subtests in the following order: attention and calculation, language, drawing, and the four subtests of the MMSE-2: BV [56].

Baek and colleagues investigated the validity of the three versions of the MMSE2 in 323 outpatients and inpatients at the Clinical Neuroscience Center at the Seoul National University Bundang Hospital who complained of memory disturbance or a decline in cognitive functioning [56]. The participants underwent a medical examination via an interview, a neurological examination, blood tests, brain imaging with CT or MRI, and neuropsychological assessments to obtain a diagnosis. MCI was diagnosed according to the Petersen criteria, and this diagnosis was supported by a CDR of 0.5. The patients with AD were diagnosed with ‘probable AD’ based on the criteria of the National Institute of Neurological and Communicative Disorders and Stroke and the Alzheimer’s Disease and Related Disorders Association (NINCDS-ADRDA) [80].

For discriminating healthy older adults from patients with MCI, the sensitivity of the MMSE-2:BV was 60%, and the specificity was 75% when using a cutoff score of ≤14 out of 16.

For discriminating healthy older adults from patients with MCI, the sensitivity of the MMSE-2:SV was 74%, and the specificity was 59% when using a cutoff score of ≤26 out of 30 [56].

## 13. Mini Addenbrooke’s Cognitive Examination (Mini-ACE/MACE)

The Mini-Addenbrooke’s Cognitive Examination is a shortened version of Addenbrooke’s Cognitive Examination-Revised (ACE-R) and Third edition (ACE-III), developed by the Mokken scaling analysis of these longer instruments. Mini-ACE consists of tests of attention, memory (7-item name and address), verbal fluency, clock drawing, and memory recall (score range 0–30, impaired to normal), and takes about 5–10 min to administer.

Larner included 755 patients between 2014–2018 new patient referrals administrating the MACE. He reported a sensitivity of 91% and a specificity of 71% at a cutoff point of ≤20 for the diagnosis of dementia. For the detection of MCI, he reported a sensitivity of 90% and a specificity of 57% at a cutoff point of ≤24. Diagnosis of dementia or mild cognitive impairment was made by the judgement of an experienced clinician using standard diagnostic criteria (DSM-IV; Petersen) [57].

## 14. Discussion

The aim of this study was to identify practical, short, and widely applicable screeners that can be used preoperatively for recognizing MCI in elderly individuals. The sensitivity and specificity range to detect MCI in an elderly population is the highest for the MoCA, with a sensitivity range of 81–93% and a specificity range of 74–89% [44]. The best screener for recognizing cognitive impairment (MCI + dementia) is the informant form of the AD8, with a sensitivity of 85% and a specificity of 86%, as reported by Galvin and colleagues [46]. In another study, the patients’ self-rating showed a sensitivity of 80% and a specificity of 59%, even at a cutoff score of 1 or below [47].

Our results add to the findings of a recent review about cognitive screening in the preoperative setting that used other inclusion criteria [82]. The number of domains of cognitive functioning was not uniformly defined, and comprehensive neuropsychological assessment as the gold standard was not routinely performed for the included screener. In particular, the latter is in our view one of the most important requirements for selecting a good screening instrument.

Although previous review articles in this research area exist [82,83,84], we wrote this review from a different perspective. Our focus is on patients with probable MCI, as these patients belong to a high-risk group that is often not previously diagnosed when arriving at the preoperative outpatient clinic, in contrast to patients with dementia. The review of Long et al. used a 2.5-min limit for the application of a screening instrument to detect such higher-risk patients. Valid and feasible screeners will be missed with such a short administration time. Therefore, after weighing up the advantages and disadvantages, we chose to limit the administration time to 15 min to still select instruments feasible to handle while having a minimum test time also fitting for older patients.

In ‘A Best Practices Guideline’ [85] it is strongly recommended to perform a cognitive assessment in patients without a known history of dementia or cognitive impairment. This guideline is based on a literature review provided by an expert panel. If cognitive impairment is detected during screening, it is worth considering further evaluation and referral to a geriatrician for further evaluation, as also advised by Roebuck-Spencer et al. [86]. Moreover, as also suggested by others, cognitive screening tests should not be used to replace comprehensive neuropsychological assessments, but rather to identify patients with subtle signs of cognitive impairment to indicate subsequent comprehensive neuropsychological evaluation. To this end, this approach using cognitive screeners can contribute to the cost-effective delivery of services, improved healthcare resource allocation, early identification of patients in need of more comprehensive diagnostic evaluation, and ultimately improved cognitive outcomes in perioperative care. Moreover, early identification of patients who may benefit from treatment becomes even more relevant when disease-modifying therapy for Alzheimer’s disease will become available [87]. However, a pitfall of the increase in cognitive screening is that tests will be taken by medical providers who typically do not have advanced training and experience with cognitive assessment or psychometrics. Research on the current most commonly used cognitive screening tests highlights that the sensitivity of these measures is lacking and indicates a need to rely on clinical risk factors in addition to screening tests when determining who should be referred for further comprehensive neuropsychological assessment [86].

For some of the studies included in this review, one or more cautionary remarks must be made. First, some studies equated a CDR of 0.5 to MCI. However, it should be noted that the CDR is a severity rating scale and not a diagnostic instrument [88]. Nevertheless, in our review, a comprehensive neuropsychological assessment was considered the gold standard. Second, it is noteworthy that in the study of Fujiwara and colleagues, a MoCA cutoff score of ≤25 best predicts both MCI and Alzheimer’s dementia [44]. Third, Molnar and colleagues investigated the O3DY screener in patients selected on the basis of their MMSE score, followed by a comprehensive neuropsychological assessment. However, some elements appear in both the O3DY and the MMSE [45]. Fourth, no women were included in the study of Tariq and colleagues about the validation of the SLUMS. This is a limitation since sex may affect the etiology, presentation, or treatment outcomes of many diseases, which is also seen in cognitive disorders [89]. For instance, women with MCI show faster cognitive and functional decline than men, as studied by Lin et al. and Sohn et al. [90,91]. In addition to sex differences, ethnic disparities exist in Mild Cognitive Impairment, as investigated by Wright et al. [92]. Hispanic and African American participants had a greater likelihood of developing MCI and dementia than Caucasian participants accounting for age and education differences. Hispanic participants had greater odds of MCI or dementia than both Caucasian and African American participants adjusting for sociodemographic variables, vascular risk factors, and brain imaging factors. On the other hand, other literature suggested that although African Americans are more likely to develop cognitive impairment, this association may be due to other socio-demographic factors than ethnicity in itself [93]. Fifth, in contrast to all other studies, two studies did not report which criteria were used to define MCI [53,54]. Finally, it should be noted that the articles selected in this review focus on screeners. Therefore, the tools presented in this review are not suitable for diagnosing dementia or mild cognitive impairment, as diagnosing MCI requires an extensive interview of the patients, a primary caregiver, and often additional neuropsychological testing. Therefore, a positive screening outcome should lead to consideration of referral for comprehensive neuropsychological evaluation and should lead to a discussion about the urgency of the surgery and the risks/benefits for the patient. If surgery has to take place in patients with cognitive decline, a strategy must be set up for the prevention of POCD and delirium after surgery and anesthesia.

In this review, the sensitivity and specificity of diagnostic tests are displayed as ranges. These ranges may possibly exist because of the differences in cutoff scores and in criteria for diagnosing MCI. For example, in the study of Freitas and colleagues [42], the ideal cutoff point of the MoCA was much lower than the original cutoff of 26 proposed by the authors. There are more recent studies that use lower cutoff scores for detecting MCI [94,95]. In a Finnish surgery population with patients older than 74, a cutoff score of 19 or lower was reported to identify patients with MCI [96], while the diagnostic accuracy of the Japanese MoCA was almost the same as that of the original version by Nasreddine and colleagues at the same cutoff score [41]. Cutoff scores can even vary because of differences in educational levels. In a study by Milani and colleagues, optimal MoCA cutoffs per race/ethnicity and education level were calculated to distinguish between normal cognition and MCI [97].

It should be mentioned that this review has its strengths and limitations.

We displayed the sensitivity and specificity of the various cognitive screeners for both mild cognitive impairment and cognitive impairment overall, if possible, while previous reviewers paid more attention to Alzheimer’s disease instead of MCI or CI, such as the systematic review of De Roeck and colleagues [84]. Because MCI is important to recognize in the preoperative setting, this is the greatest strength of our review. It is time to routinely screen elderly surgical patients preoperatively for the presence of cognitive impairment [98].

The first limitation is that we did not conduct a detailed quality rating of each individual article included in this review. By using strict inclusion and exclusion criteria, we tried to minimize this limitation as much as possible, which fits with the narrative perspective of this review. The second limitation is that, with the selected screeners, we focused on the detection of amnestic MCI by setting the exclusion criteria that the screener at least assesses the cognitive domain memory. Therefore, other subtypes of MCI, such as non-amnestic or single domains, can be missed. We added this criterion, as most types of MCI include problems with memory. However, the majority of the included articles did not mention this subgroup or grouped all subtypes into one diagnostic MCI group. The last limitation is that some screeners can be used as a decision tree by combining components of different screeners, such as the MiniCog and CODEX. We excluded the CODEX as it is overlapping the components of the Mini-Cog too much and more evidence about Mini-Cog does exist than about CODEX.

New technologies and platforms such as computer tablets offer many opportunities for creating tasks and interactive experiences from which cognitive status could potentially be inferred [99]. Serious games have been promoted as a way to stimulate cognitive activity in elderly users, improve brain fitness, or preserve cognitive status using specially designed training games, many of which are intended to enhance or preserve working memory [100]. However, to the best of our knowledge, there are no valid and reliable gamified tools to monitor cognition perioperatively available yet.

Future studies are needed to validate cognitive screeners prospectively in the preoperative setting to identify patients at risk for postoperative delirium and postoperatively assess cognitive dysfunction. In the context of making screening less burdensome and motivating patients to monitor their cognitive performances before and after surgery, it would be interesting to perform a study with a gamified screener.

## 15. Conclusions

According to our criteria, the MoCA has the best practical features and most effective diagnostic properties to identify patients with mild cognitive impairment as a preoperative cognitive screener. With this screener, routine screening of presurgical cognitive function can be both feasible and valid. This warrants further implementation and research in surgical pathways. More knowledge about preoperative cognitive status may help to prevent or treat postoperative cognitive dysfunction and to diminish its burden on many older surgical patients.

## Figures and Tables

**Figure 1 healthcare-10-01112-f001:**
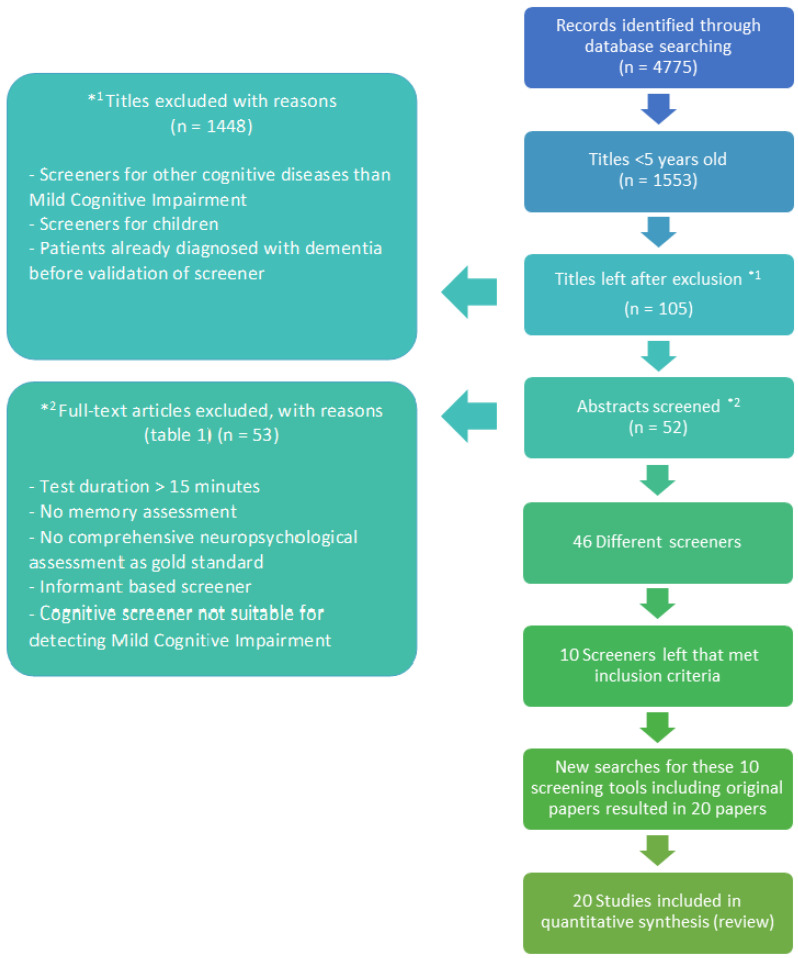
Flowchart showing the search strategy and article/screener selection after the literature and snowball search.

**Table 1 healthcare-10-01112-t001:** This table shows all cognitive screeners that were excluded during the search process because they did not match the inclusion criteria, or they did match the exclusion criteria.

Screener	Exclusion Criterion
Informant Questionnaire on Cognitive Decline in the Elderly (IQCODE)	<2 domains of cognitive functioning and informant based screener
NeuroCogFX	Test duration > 15 min
National Institutes of Health Toolbox Cognitive Battery (NIHTB-CB)	Test duration > 15 min
Short Blessed Test (SBT)	No articles with comprehensive neuropsychological assessment as gold standard for detecting MCI
National Institutes of Health Toolbox Cognitive Battery (AMNART)	<2 domains of cognitive functioning No memory assessment
Auditory Verbal Learning Test (AVLT)	Test duration > 15 min
Clock drawing test (CDT)	No memory assessment
Clock in the box	No memory assessment
Cognitive Disorder Examination (CODEX)	Not a real test, it is a decision tree of the Mini-Cog/too much overlap with Mini-Cog
Cognitive Activity Scale	No memory assessment
Controlled Oral Word Association Test (COWAT)	No memory assessment
Verbal fluency test	No memory assessment
DemTect	No articles with comprehensive neuropsychological assessment as gold standard for detecting MCI
Hasegawa Dementia Scale (HDS)	Screener for delirium
Identification of Seniors At Risk (ISAR)	No articles with comprehensive neuropsychological assessment as gold standard for detecting MCI
Stroop color word test (SCWT)	No memory assessment
Addenbrooke’s Cognitive Examination (ACE)	Test duration > 15 min
Animal fluency test	No articles with comprehensive neuropsychological assessment as gold standard for detecting MCI
Brief Screen Cognitive Impairment (BSCI)	No articles with comprehensive neuropsychological assessment as gold standard for detecting MCI
Geriatric 8 (G8)	No memory assessment
Mail-in Cognitive Function Screening Instrument (MCFSI)	No memory assessment and study partner or informant is needed
Month Backward Test (MBT)	No articles with comprehensive neuropsychological assessment as gold standard for detecting MCI
Time & Change	No articles with comprehensive neuropsychological assessment as gold standard for detecting MCI
Trail making A&B	No memory assessment
Brief Neuropsychological Battery (BNB)	Test duration > 15 min
Cognitive Performance Scale (CPS)	No articles with comprehensive neuropsychological assessment as gold standard for detecting MCI
Literacy Independent Cognitive Assessment (LICA)	Test duration > 15 min and no articles with comprehensive neuropsychological assessment as gold standard for detecting MCI
Memory Fluency and Orientation (MEFO)	No articles with comprehensive neuropsychological assessment as gold standard for detecting MCI
Rapid Cognitive Screen (RCS)	No articles with comprehensive neuropsychological assessment as gold standard for detecting MCI
Computerized Assessment of Mild Cognitive Impairment (CAMCI)	Test duration > 15 min
Short Portable Mental Status Questionnaire (SPMSQ)	No articles with comprehensive neuropsychological assessment as gold standard for detecting MCI
The 5 object test	No articles with comprehensive neuropsychological assessment as gold standard for detecting MCI
Brief Memory and Executive Test (BMET)	No articles with comprehensive neuropsychological assessment as gold standard for detecting MCI
Dementia Rating Scale 2	Test duration > 15 min
Frontal Assessment Battery (FAB)	Specific for frontal lobe dysfunction
Cogstate Brief Battery (CBB)	Specific for nondementia brain injuries

**Table 2 healthcare-10-01112-t002:** This table shows a summary and comparison of the included cognitive screeners and the available literature. MCI = Mild Cognitive Impairment, CI = Cognitive Impairment (MCI + Alzheimer Disease), NR = Not Reported. * Cut off ≤23.5 for a population with less than high school education. Cut off <25.5 for a population with higher education. ** Only applicable to the modified TICS (TICS-M).

Tool	Items/Cognitive Domains	Author/Setting Recruitment	*N*	Average Age (years)	Admin Time (min)	TP/Cutoff Score	Blinding Index Test/Reference Test	Sensitivity + Specificity MCI	Sensitivity + Specificity CI
**Mini-Cog**	- Three word recall - Clock drawing test	Carnero-Pardo and colleagues [39]/Primary care Madrid and Granada	307	All 72	3	5/≤1	Yes	-	Sen 60% Spe 90%
		Li and colleagues [40]/Neurological outpatient department Cangzhou City Central Hospital	229	MCI 68.7 Non-MCI 66.1		5/≤1	NR	Sen 85.71% Spe 79.41%	-
**MoCA**	- Recall - CDT - Trail making - Orientation	Nasreddine and colleagues [41]/Jewish General Hospital (JGH) Memory Clinic in Montreal and University of Sherbrooke NRS memory clinic	277	NC 72.84 MCI 75.19 Dementia 76.72	10	30/≤25	NR	Sen 90% Spe 87%	-
		Freitas and colleagues [42]/Dementia Clinic, Neurology Department of the Coimbra University Hospital	360	NC 71.34 MCI 70.52 Dementia 74.22		30/≤21	NR	Sen 81% Spe 77%	-
		Sokolowska and colleagues [43]/Department of Geriatrics, Collegium Medicum in Bydgoszcz, Nicolaus Copernicus University, Torun	131	MCI 79.06 Non-MCI 74.8		30/≤24	Yes	Sen 89.5% Spe 74.1%	-
		Fujiwara and colleagues [44]/Memory Clinic of Tokyo Metropolitan Geriatric Hospital, Tokyo	96	NC 76.4 MCI 77.3 Dementia 77.5		30/≤25	Yes	Sen 93% Spe 89%	-
**O3DY**	- Day - Date - WORLD spelled backward - Year	Molnar and colleagues [45]/randomly selected samples throughout Canada	1560	All 79.5	2–3	4/≤3	Yes	-	Sen 80% Spe 56%
**AD8**	- 12 yes/no questions about memory, problem-solving, orientation, etc.	Galvin and colleagues [46]/Community-dwelling volunteers who enrolled in a longitudinal study of healthy aging and dementia.	236	All 78.1	2–3	8/≤1	Yes	Sen 74% Spe 86%	Sen 85% Spe 86%
		Galvin and colleagues [47]/Community-dwelling volunteers who enrolled in a longitudinal study of healthy aging and dementia.	325	All 76.8		8/<1	Yes	-	Sen 80% Spe 59%
**SAGE**	- Orientation - Naming pictures - Similarities - Calculations - Memory - Construction 3D - Clock drawing - Verbal fluency - Executive function - Memory	Scharre and colleagues [48]/geriatric outpatient clinics, educational talks to lay public, independent and assisted living facilities, senior centers, free memory screens through newspaper advertisement, and Memory Disorders Clinic at Ohio State University.	63	All 78.0	15	22/≤16	Yes	Sen 62% Spe 95%	Sen 79% Spe 95%
		Scharre and colleagues [49]/educational talks to lay public, independent and assisted living facilities, senior centers, free memory screens, or at the Memory Disorders Clinic at The Ohio State University.	66	All 75.2		22/≤15	Yes	Sen 69% Spe 86%	Sen 71% Spe 90%
**SLUMS**	- Orientation - Short-term memory - Calculations - Naming animals - Clock drawing - Recognition of geometric figures	Tariq and colleagues [15]/Geriatric Research Education and Clinical Center (GRECC), Veterans’ Affairs Medical Center (VAMC) hospitals in Saint Louis.	702	All 75.3	7	30/≤23.5 * 30/≤25.5 *	NR	Sen 92% Spe 81% Sen 95% Spe 76%	- -
		Shwartz and colleagues [50]/Mild Cognitive Impairment (MCI) Clinic at the Atlanta Veterans Affairs Medical Center (VAMC).	148	All 68.48		30/≤25	No	Sen 81% Spe 68%	-
**TICS-M**	- Orientation - Repetition - Naming - Attention - Calculation - Immediate and delayed recall **	Cook and colleagues [51]/Community-dwelling older adults. Articles in a local senior newspaper, advertisements in the community, and from the participant pools of other local aging investigators.	71	All 74.9 167	10	50/≤34	Yes	Sen 82.4% Spe 87.0%	-
		Knopman and colleagues [52]/Mayo Clinic Study of Aging and the Mayo Clinic Alzheimer’s Disease Research Center (ADRC).	167	NC 81 MCI 84 Dementia 80		50/≤31	Yes	Sen 82.4% Spe 87.0%	-
**Quick-MCI**	- Orientation - Registration - Clock drawing - Delayed recall - Verbal fluency - Logical memory	O’Caoimh and colleagues [53]/four memory clinics across Ontario, Canada (Hamilton, Paris, Niagara Falls, and Grand Bend).	965	NC 67 MCI 75.5 Dementia 79	3–5	100/NR	Yes	Sen 82% Spe 70%	Sen 91% Spe 80%
		Bunt and colleagues [54]/a geriatric outpatient department in a regional hospital, in the North of the Netherlands.	90	NC 68.7 MCI 79.1 Dementia 79.2		100/≤51.5	Yes	Sen 82% Spe 90%	-
		Glynn and colleagues [55]/electronic journal databases EBSCO, Psych info, and Science Direct.	NR	NR		NR	Yes	Sen 82% Spe 82%	Sen 95% Spe 83%
**MMSE2 BV**	- Registration - Orientation - Recall - Attention - Calculation - Language - Drawing - Registration - Orientation - Recall	Baek and colleagues [56]/outpatients and inpatients at the Clinical Neuroscience Center at the Seoul National University Bundang Hospital.	414	NC 67.05 MCI 71.05 Dementia 75.38	5	16/≤14	NR	Sen 60% Spe 75%	-
**MMSE2 SV**					10–15	30/≤26	NR	Sen 74% Spe 59%	-
**Mini-ACE**	- Orientation - Memory - Language - Visuospatial function	Larner [57]/Cognitive Function Clinic, Walton Centre for Neurology and Neurosurgery, Liverpool.	755	All 60	5–10	30/≤25	NR	-	Sen 91% Spe 71%
						30/≤24		Sen 90% Spe 57%	-

**Table 3 healthcare-10-01112-t003:** This table shows the interrater reliability and the test-retest reliability of the included screeners.

	Interrater Reliability (Correlation)	Test-Retest Reliability (Correlation)
**Mini Cog**	0.95 [58]	-
**MoCA**	0.852 [59]	0.92 [41]
**MMSE-2 BV**	0.94–0.99 [56]	0.76 [56]
**MMSE-2 SV**	0.94–0.99 [56]	0.82 [56]
**TICS-M**	0.90 [60]	0.91–0.95 [60,61]
**SAGE**	0.96 [48]	0.86 [48]
**SLUMS**	-	0.82 [62]
**Quick MCI**	1.00 [63]	0.86–0.87 [63,64]
**AD8**	0.80–0.89 [65,66]	0.67–0.81 [65,66]
**O3DY**	0.64 [67]	-
**Mini-ACE**	-	0.64 [68]

## Data Availability

Not applicable.

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
