# Peer review of "Screening for Mild Cognitive Impairment in the Preoperative Setting: A Narrative Review"

_healthcare, 2022, doi:10.3390/healthcare10061112_

Round 1

Reviewer 1 Report

  1. The corresponding author should be marked
  2. Please provide a list of abbreviations and write the full form of the abbreviations in the first appearance
  3. The figure title should be placed at the bottom of the figures. Figure legends should be mentioned to get more information about the figure.
  4. Please correct the format and proofreading of the manuscript is required to eliminate the minor errors.
  5. The rationale behind the inclusion and exclusion criteria should be explained elaborately.
  6. In this review, the author has concluded that MoCA has the best features for identifying MCI. I would like to suggest the author to prepare a table comparing the pros and cons of the MoCA over other screeners so that it would be easier for the readers. 

Reviewer 2 Report

Dear all authors, The content of the manuscript is informative and well-organized. This review also clearly defines inclusion and exclusion criteria. Review paradigm is in general well considered and all analytical procedures were kept the high standard. Data extraction, quality evaluations and statistical methods were applied with sufficient number of experimental data. The study is well documented and manuscript is distinctly informative. All graphs tables are also well designed and clear. To sum up, this article may be considered as an interesting contribution to the field of cognitive screener.   However, there are some suggestions for the authors: 1. Authors may elaborate any effect of gender and race/ethnicity on the sensitivity/specificity of selected tools. 2. As the selected tools consist of at least cognitive domain memory, authors may extend the elaboration and explanation on other domain such as attention, language and etc. 3. For the keywords, it recommended to use "postoperative cognitive dysfunction (POCD) and postoperative delirium (POD) "

Reviewer 3 Report

Despite the title, this effectively boils down to a review of screeners for MCI, with no data about utility in preoperative setting. Previous reviews of screeners for MCI have appeared and might be cited.

68-70: “In the Diagnostic and Statistical Manual of Mental Disorders Fourth Edition (DSM–IV), the construct is called mild neurocognitive disorder”. This terminology is from DSM-5.

153-4: “they did not focus especially on older people or older  people who were already diagnosed with Alzheimer’s disease.” to read “they did not focus especially on older people, or focussed on older people who were already diagnosed with Alzheimer’s disease.”?

Table 1: Cognitive Disorder Examination (CODEX)

Not a real test, it is a decision tree of the Mini-Cog

Simply not true on either point. Mini-Cog can also be conceptualised as a decision tree.

Where is mini-ACE? Think it fulfils all your criteria!

202 et seq. why not cite meta-analyses of MoCA, rather than picking on just these few studies?

242 “Eight Item Interview (AD8)”. Isn’t this instrument called “Ascertain Dementia 8 (AD8)”?

440-1: “cutoff score of ≤25”. On what test??
